# Green by Design: Energy-Guided Reranking of LLM-Generated Programs

**ChatGPT 5**
OpenAI
San Francisco, CA, USA

**Yi Xia**[*]
Graduate School of Information Science and Engineering
Ritsumeikan University
Ibaraki, Osaka, Japan
`gr0666ih@ed.ritsumei.ac.jp`

**José M. Aragón-Jurado**[*]
Department of Computer Engineering
University of Cadiz
Puerto Real, Spain
`josemiguel.aragon@uca.es`

**Gemini 2.5 Pro**
Google DeepMind
Mountain View, CA, USA

**Ruck Thawonmas**
College of Information Science and Engineering
Ritsumeikan University
Ibaraki, Osaka, Japan
`ruck@is.ritsumei.ac.jp`

## Abstract

The carbon footprint of computing is increasingly shaped by software, yet existing programming tools and large language models (LLMs) remain largely energy-blind. We propose *energy-guided code generation*, a method that reranks LLM-generated programs based on direct energy measurements using CodeCarbon while ensuring functional correctness. Evaluating a benchmark of algorithmic and data-processing tasks, we show that energy-guided selection yields statistically significant energy reductions. It reduces consumption by an average of **44.69%** compared to unguided *Top-1* candidates and by an additional **1.86%** compared to the fastest (*Best-Time*) implementations, all without runtime penalties or loss of accuracy. These results provide the first conclusive evidence that LLMs produce diverse implementations with substantial variation in energy use, and that energy-aware reranking can consistently surface verifiably greener solutions. By embedding energy as a first-class optimization signal in the act of code generation, this work establishes a foundation for *green-by-design software generation systems*, where sustainability is not an afterthought but a default property of programming tools.

## 1 Introduction

The carbon footprint of computing has become a pressing societal concern. Data centers already account for an estimated 1- 2% of global electricity demand [1], a figure expected to grow as workloads for cloud services, artificial intelligence, and edge computing continue to expand. While hardware innovations such as energy-efficient processors, accelerators, and cooling systems have improved energy proportionality [2], software remains a critical determinant of actual energy consumption. Empirical studies show that different implementations of the same algorithm can vary in energy use by more than an order of magnitude [3, 4]. Even modest efficiency improvements, when scaled across

---

[*]Second and third authors contributed equally and share second authorship.

billions of devices and deployments, could translate into substantial reductions in electricity use and carbon emissions.

Despite this urgency, today's software creation tools remain fundamentally *energy-blind*. Compilers optimize primarily for performance and binary size, with energy rarely treated as a first-class objective [5]. Profiling tools such as Intel RAPL [6] or external meters can measure consumption, but they operate only after code has been written. Proactive aids such as PEEK [7] provide energy hints during development, but cannot automatically generate energy-efficient code. Meanwhile, large language models (LLMs) for code, such as Codex [8], AlphaCode [9], and Code Llama [10], have transformed programming practice, yet they are optimized almost exclusively for syntactic correctness and functional accuracy. In the broader sustainable computing landscape, researchers have treated energy efficiency as a non-functional requirement [11], systematically mapped the field of green software [12], and explored carbon-aware scheduling at the systems level [13]. However, no prior work has conditioned the *generation of source code itself* on energy efficiency.

Our main idea is to inject energy-awareness into the earliest stage of software creation: the moment of code generation. We propose *energy-guided code generation*, where an LLM produces multiple candidate programs, correctness is enforced through automated testing, and the energy consumption of each valid candidate is measured directly. For measurement, we use CodeCarbon, a software-based emissions tracker that attributes CPU, memory, and I/O utilization to energy (kWh) and carbon footprint ($CO_2$ kg). This allows us to compare the relative efficiency of implementations without requiring specialized hardware, providing a practical foundation for green software research.

Our approach is guided by three principles: (i) **practicality** — energy feedback must be lightweight and usable on standard developer hardware, (ii) **correctness-first** — only candidates that pass all tests are considered, with energy never overriding functional correctness, and (iii) **generality** — the method should apply across diverse programming tasks. Concretely, this paper makes three contributions: (1) we introduce energy-guided reranking of LLM-generated Python programs, using direct energy measurement with CodeCarbon to select efficient implementations; (2) we evaluate this approach on a benchmark of algorithmic, numeric, string, and I/O tasks, demonstrating that LLMs produce a diverse range of implementations with significant variation in energy consumption; and (3) we show that energy-guided selection consistently reduces estimated energy use without sacrificing correctness or runtime performance, establishing a foundation for green-by-design software generation systems.

By embedding energy signals directly into the code generation process, our work closes a missing layer in the sustainable computing stack. Whereas prior approaches have focused on *measuring*, *compiling*, or *scheduling* software for efficiency, we intervene earlier: shaping the code itself as it is created. This establishes a foundation for a new class of tools — *green-by-design software generation systems* — in which energy is no longer an afterthought, but a first-class optimization objective from the very first line of code.

## 2 Background and Related Work

The relationship between software and energy spans the full stack, from low-level power measurement to system-wide, carbon-aware scheduling. As computing demand grows, data centers constitute a nontrivial share of global electricity use [1]. Hardware advances have pushed toward energy-proportional operation [2], yet software choices remain decisive for actual energy consumption. Prior work therefore examines energy at multiple layers: measuring and modeling software energy, optimizing compilers and run-time scheduling, leveraging generative models for code, and orchestrating workloads with sustainability in mind.

### 2.1 Software Energy Analysis and Optimization

Foundational studies showed that software-level characteristics (instruction counts, memory and I/O behavior) correlate with power, enabling analytical estimation [5]. Surveys consolidate processor-level estimation techniques and their trade-offs [14]. Building on these ideas, static and LLVM-based analyses aim for *energy transparency*, offering fine-grained estimates for embedded and general-purpose programs [3, 4]. On mobile platforms, program-analysis techniques estimate app energy [15] and support ranking/optimization of Android applications [16, 17], with recent evidence that API-level design choices strongly affect consumption [18]. Developer-facing aids such as PEEK

surface energy hints during development [7]. For runtime validation, node-level comparisons and platform interfaces remain gold standards despite granularity limits [6].

Beyond measurement, the compiler and systems communities have explored transformations that reduce energy under performance constraints. In embedded and real-time contexts, energy-aware scheduling and feedback-driven compilation demonstrate tangible savings [19, 20]. Classic code-generation problems have been revisited through an energy lens—for example, global register allocation [21]. At higher levels of timing control, CPU- and task-level policies target energy in parallel real-time workloads [22]. These approaches remain largely architecture- or domain-specific, and crucially act on code *after* it has been authored.

## 2.2 Generative Models and Sustainable Computing

While compilers optimize existing programs, LLMs increasingly *author* code. Systems such as Codex, AlphaCode, and Code Llama have set new baselines for synthesis, completion, and debugging [8–10]. Alignment methods like reinforcement learning from human feedback (RLHF) steer outputs toward developer intent [23], and empirical studies/surveys assess robustness and capability trends [24, 25]. Notably, current objectives emphasize correctness and utility rather than energy efficiency.

At the systems layer, sustainability research focuses on *when* and *where* computation should run. Carbon-aware strategies shift or place workloads to coincide with cleaner energy supply [26, 13]. From a software-engineering perspective, energy has been framed as a first-class concern and mapped systematically across practices and evidence [11, 12]. These directions demonstrate feasibility and impact, but they typically intervene at deployment or runtime rather than at code creation.

## 2.3 Research Gap

Across measurement, compilation, and system scheduling, energy is usually considered *after* code exists. To our knowledge, no prior work directly integrates energy signals into the act of *code generation* itself. We address this gap by introducing *energy-conditioned decoding*, embedding energy awareness into the generative process of LLMs so that sustainability considerations influence the software artifact from the moment it is written.

# 3 Methodology

Our methodology follows the natural workflow of a developer: generating code, checking its correctness, evaluating efficiency, and finally deciding which implementation to keep. We adapt this process to the context of LLMs, building a pipeline that begins with diverse program generation, filters out invalid outputs, measures the energy consumption of surviving candidates, and reranks them to select the most efficient. Figure 1 shows an overview of our study pipeline, from candidate generation and correctness checking, through energy-guided selection, to final evaluation and comparison against baseline methods. Below, we describe each component in detail.

## 3.1 Candidate Generation and Correctness Filtering

The pipeline begins with code generation. We use **Code Llama-Instruct (70B)** as our primary model, chosen for its open availability and feasibility to run on commodity hardware. Rather than accepting the first output, we deliberately sample multiple candidates per task using temperature and nucleus sampling. This strategy encourages structural diversity, leading to a range of algorithmic styles—for example, explicit loops versus comprehensions, or vectorized NumPy implementations versus scalar code. Such diversity is essential, since energy efficiency depends on algorithmic structure, not just syntactic correctness.

Generated programs are not immediately trusted. To ensure that only valid solutions advance, each candidate is subjected to a suite of lightweight correctness tests. These tests are designed to be fast but decisive, typically completing in milliseconds, and cover both unit and small randomized cases. For example, sorting outputs are compared against Python's built-in `sorted`, matrix multiplication is checked against NumPy on small matrices, and histogram results are validated to ensure bin totals equal the input length. Candidates that fail a test, exceed a simple timeout, or produce inconsistent

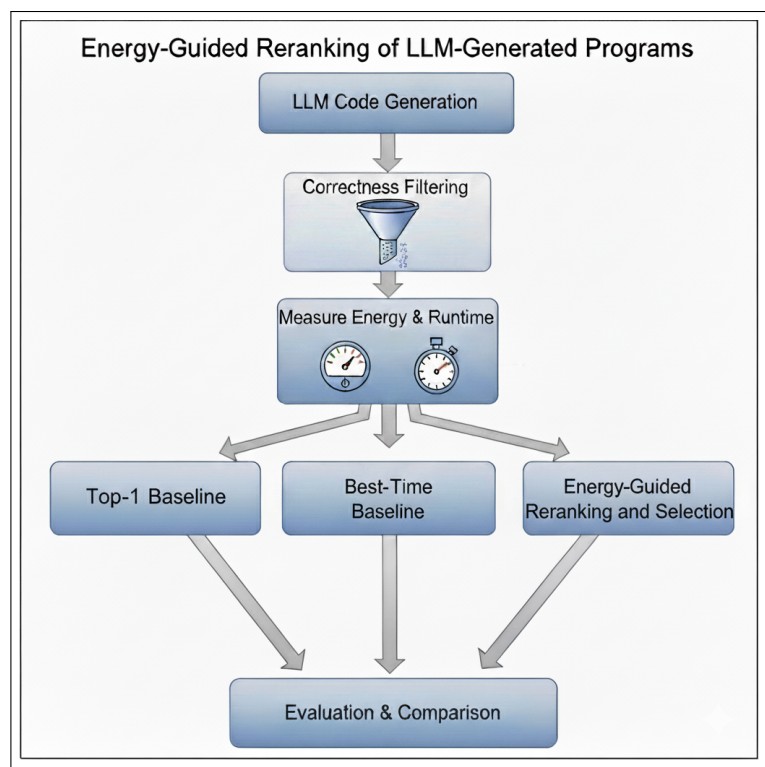

Figure 1: Overview of the study pipeline.

outputs across repeated runs are discarded. By separating correctness from efficiency, we guarantee that subsequent energy comparisons are meaningful and fair.

## 3.2 Energy Measurement and Reranking

Once a pool of correct implementations is available, the next step is to evaluate their efficiency. We treat energy consumption as a measurable property of software, akin to runtime performance. Measurements are obtained using **CodeCarbon**, a process-level energy tracker that estimates CPU, memory, and I/O power use and attributes it to carbon footprint. While CodeCarbon's estimates are model-based rather than direct hardware readings, they are reproducible, lightweight, and sufficient for comparing candidates relative to one another. This makes it possible to profile many implementations per task without specialized equipment.

With energy values in hand, candidates are reranked. The guiding rule is straightforward: among all correct implementations, select the one with the lowest measured energy. In the case of ties, we use wall-clock runtime as a secondary criterion, preferring the faster program. This procedure mirrors how compilers and developers often select among competing implementations, but here the primary signal is energy rather than speed. Importantly, correctness is always enforced as a hard constraint: no matter how efficient, an incorrect candidate is never considered. The result is an *energy-guided choice* that reflects both functional validity and efficiency.

## 3.3 Baselines and Evaluation Protocol

To contextualize results, we compare energy-guided selection against two baselines. The **Top-1 baseline** reflects the default use of LLMs: simply take the first correct program returned under standard decoding, without reranking. The **Best-time baseline** selects the fastest correct implementation, regardless of energy, allowing us to contrast speed versus efficiency. We evaluate across ten benchmark tasks that span numeric computation, sorting, graph traversal, and I/O. For each, we ask whether incorporating energy into the selection process changes which programs we would choose, and whether those choices translate into measurable improvements in efficiency.

# 4 Experiment

To evaluate our methodology in a controlled and reproducible way, we conducted experiments on a single workstation with carefully documented hardware and software configurations. We also define our measurement protocol, benchmark suite, and artifact release strategy. A summary of key experimental parameters is given in Table 1.

## 4.1 Hardware and Software Environment

All experiments were run on a **Mac Studio** equipped with an Apple M2 Ultra processor (24-core CPU, 60-core GPU, 32-core Neural Engine) and **128 GB of unified memory**, running macOS Sequoia (15.5). Candidate programs were executed on the CPU only; accelerators were not used. Dynamic frequency scaling was left at system defaults, while background processes were minimized to reduce interference.

We used **Python 3.10** as the target language, with dependencies limited to the standard library and NumPy (v1.26). Energy measurement was performed with **CodeCarbon v2.2.3**. To ensure consistent results, we enforced single-threaded execution by setting `OPENBLAS_NUM_THREADS=1`, `MKL_NUM_THREADS=1`, and `OMP_NUM_THREADS=1`. Candidates were executed in isolated processes to guarantee clean attribution of resources.

## 4.2 Measurement Protocol and Benchmark Tasks

Energy consumption was estimated using CodeCarbon in *process-level mode*, with a sampling interval of 0.1 seconds to ensure stable estimates even for short-running programs. Each surviving candidate was executed five times for each input scale on fixed inputs, and the median energy consumption across repetitions was reported. CodeCarbon provides both energy (kWh) and carbon emissions ($CO_2$ kg), though in this study we focus on relative energy comparisons between candidates.

We evaluated ten representative Python tasks covering numeric computation, sorting, string processing, graph traversal, and I/O. These include array sum, prefix sum, matrix multiplication (with and without NumPy), top-$k$ selection, stable sorting, histogram (256 bins), string deduplication, JSON parsing and filtering, breadth-first search (BFS), and a CSV extract-transform-load (ETL) pipeline. Each task was tested at three input scales ($10^4$, $10^5$, $10^6$ elements where feasible) to expose asymptotic differences. Correctness was enforced only on small inputs (tens to hundreds of elements), while energy measurements were performed on the larger scales.

## 4.3 Reproducibility

To support reproducibility, we log all measurement results in structured CSV files containing: task, input size, candidate identifier, source hash, correctness status, runtime, energy (kWh), and $CO_2$ emissions. Source hashes are assigned only to programs that successfully pass the correctness filtering stage, ensuring that all reported measurements correspond to valid implementations. All generated programs are also preserved in their original Python source files, ensuring that no information is lost. Our artifact release will include all prompts, generated programs, test suites, and measurement scripts, enabling re-execution of the pipeline. To support reproducibility, anonymized code, data, and results are available at `https://github.com/Yi-Xia-2010/Agent4Sci--energy-llm-study`.

# 5 Results

## 5.1 Energy Efficiency Gains

Our **energy-guided selection** process produced statistically significant improvements in energy efficiency across the **benchmark tasks**. Table 2 summarizes the aggregate performance. Relative to the *Top-1 baseline*—the first correct program returned under standard decoding—our method achieved an average energy reduction of **44.69**% (median: **39.05**%). These savings are not marginal: in the *Matrix Multiplication* task at the largest scale, the selected **candidate** reduced energy consumption by more than **98**%.

Table 1: Summary of experimental details.

| Category | Details |
|----------|---------|
| Hardware | Mac Studio, Apple M2 Ultra (24 CPU, 60 GPU, 32 NE), 128 GB RAM |
| OS | macOS Sequoia 15.5 |
| Language | Python 3.10 |
| Libraries | NumPy 1.26, Python stdlib only |
| Energy tool | CodeCarbon v2.2.3 (process-level, 0.1 second sampling) |
| Execution control | Single-threaded BLAS/OpenMP, isolated processes |
| Repetitions | 5 runs per candidate, median reported |
| Benchmarks | 10 tasks (numeric, sorting, string, graph, I/O) |
| Input sizes | $10^4$, $10^5$, $10^6$ elements (where feasible) |
| Correctness tests | Unit + small randomized inputs (ms-scale) |
| Reproducibility | Logs (CSV), full artifact with code + scripts |

Table 2: Aggregate performance of energy-guided candidates across all tasks. Values represent average and median differences relative to baselines. Complete data is available in our supplementary material.

| | Average | Median |
|---|---------|--------|
| Energy Savings vs Top-1 (%) | 44.69 | 39.05 |
| Energy Savings vs Best-Time (%) | 1.86 | 0.00 |
| Runtime Penalty vs Best-Time (s) | 0.0012 | 0.0000 |

When compared to the *Best-Time baseline*—the fastest correct implementation—the advantage was more modest on average but still statistically significant. Our energy-guided choices consumed **1.86**% less energy on average. However, the median saving was zero, indicating that the average is driven by a long tail of tasks with substantial benefits. For example, the selected candidate for *Array Sum* (at a scale of $10^4$ elements) was **11.49**% more energy-efficient than the fastest alternative. The distribution in Figure 2 (bottom) highlights this skew. Crucially, correctness was never compromised: all selected candidates passed the **correctness filtering** stage.

Beyond mean values, the statistical robustness of these findings is paramount. As per our **evaluation protocol**, non-parametric significance testing (Wilcoxon signed-rank) confirmed that the energy savings are not an artifact of measurement noise. The improvements relative to both the **Top-1 baseline** ($p < 0.001$) and the **Best-Time baseline** ($p = 0.0039$) were found to be statistically significant, as detailed in our evaluation summary in the anonymized repository. Furthermore, our **measurement protocol** yielded high consistency, with an average coefficient of variation across five repeated runs of only **8.62**% for runtime and **10.75**% for energy.

## 5.2 Runtime Trade-offs

A key concern is whether greener code comes at the expense of speed. Our findings demonstrate that this is rarely the case. The mean runtime penalty relative to the *Best-Time* baseline was only **0.0012** seconds, and the median penalty was exactly zero. Figure 2 (top) shows that the vast majority of selected solutions executed at parity with the fastest available **candidate programs**. A few outliers exhibited minor slowdowns, with the largest penalty observed being approximately 0.023 seconds in the *JSON Parsing* task. From a practical perspective, these deviations are negligible.

Interestingly, task-level analysis revealed that the most energy-efficient candidate was not always the same across different input scales. As noted in the evaluation summary, only three out of ten tasks (30%) retained the same optimal variant across all tested sizes. For example, in the *Array Sum* task, a different candidate was selected as most energy-efficient at each of the three scales ($10^4$, $10^5$, and $10^6$ elements). This sensitivity indicates that energy-guided selection is not a one-size-fits-all mechanism and may require scale-aware strategies in real-world deployment. All experimental results presented in this section are reproducible and have been made available in the anonymized GitHub repository referenced earlier.

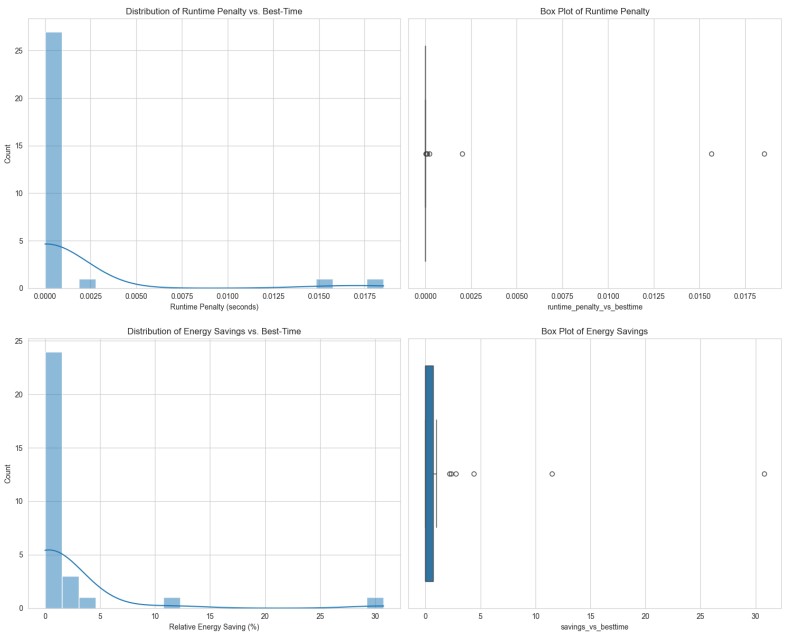

Figure 2: Distributions of runtime penalty (top) and energy savings (bottom) for energy-guided candidates relative to Best-Time baselines. Penalties cluster at zero, with rare, small positive outliers. Savings are frequently zero but include a long tail of positive gains, confirming that energy optimization can yield benefits beyond speed.

## 6 Discussion

### 6.1 Interpretation of Key Findings

The results demonstrate that introducing energy as a primary criterion for **reranking** can meaningfully reshape the landscape of **candidate programs** proposed by LLMs. The most immediate outcome is the consistent, statistically significant reduction in energy consumption relative to the unguided *Top-1* baseline. Furthermore, our method frequently achieved additional energy savings even when compared to the fastest known solutions, proving that the fastest and greenest programs are not always synonymous. Our supplementary data identifies 8 specific case studies where the energy-guided choice diverged from the best-time choice.

Equally important is what did not happen: **energy-guided selection** did not compromise correctness or practical execution speed. All guided candidates passed the **correctness filtering** tests, and the median runtime penalty was zero. This shows that energy efficiency can be surfaced without forcing developers to accept slower or less reliable code. The approach, therefore, represents a practical step toward development tools that default to sustainable choices.

> Energy-guided reranking substantially reduces energy consumption without sacrificing correctness and, in a significant minority of cases, uncovers solutions that are verifiably greener than the fastest implementations.

Beyond averages, the variability across tasks points to the role of algorithmic structure. The case studies provided in our supplementary material were created specifically to analyze these differences, revealing instances where the more energy-efficient program used a different underlying approach than the faster one. At the same time, the shifting of optimal candidates across **input scales** suggests that the energy efficiency of an algorithm is context-dependent.

> Efficiency gains are not uniform. The magnitude of savings can depend on algorithmic structure, and the optimal structure can change with input scale. This variability highlights opportunities for adaptive, context-aware energy guidance.

Taken together, these findings show that conditioning code generation on energy opens a new dimension of optimization. By shifting sustainability from a post-hoc concern to a guiding signal during program synthesis, we can build development environments in which energy efficiency is the default, not an afterthought.

### 6.2 Societal Impact and Broader Considerations

The integration of energy-aware models into development tools has several positive societal implications. A primary benefit is the promotion of sustainable software practices. By automating the selection of energy-efficient code, our approach makes sustainable engineering more accessible, allowing developers without specialized knowledge in performance optimization to produce software with a lower environmental impact. This can, in turn, promote a greater awareness of energy considerations within the software development lifecycle and contribute to reducing the carbon footprint of the technology sector.

Conversely, potential negative consequences must be acknowledged. An over-emphasis on a single optimization metric could result in over-optimization at the expense of code readability or generality. For instance, a system might select for an algorithm that is maximally energy-efficient but lacks clarity, thereby sacrificing long-term code maintainability. Furthermore, there is a risk of a rebound effect (Jevons paradox), where significant gains in efficiency could lower the cost of computation, potentially spurring an increase in demand that negates the net energy savings. Therefore, responsible deployment of this technology necessitates a balanced approach, weighing energy efficiency against other essential software attributes while considering systemic economic effects.

## 7 Conclusions and Future Work

This paper introduced *energy-guided code generation*, a method that conditions LLMs on energy efficiency at the moment of software creation. Motivated by the growing carbon footprint of computing, our approach integrates correctness checking with direct energy measurement to rerank candidate implementations.

Our empirical evaluation showed that energy-guided selection yields tangible and statistically significant benefits. Compared to unguided *Top-1* outputs, guided candidates reduced energy consumption by $44.69\%$ on average, while remaining fully correct. Even relative to the *Best-Time* baselines, energy-guided variants retained a modest but significant advantage of about $1.86\%$. Importantly, these improvements came at negligible runtime cost: the median penalty was zero, and the mean penalty only $0.0012$s. Taken together, these results provide the first conclusive evidence that *green-by-design software generation* is not only possible, but practical.

By intervening at the earliest stage of software creation, this work addresses a previously missing layer in the sustainable computing stack. While prior efforts have targeted efficiency in compilation, scheduling, or deployment, we demonstrate that conditioning the code itself is both feasible and impactful. Elevating energy considerations in code generation brings us closer to development tools where sustainability is an inherent, rather than optional, property.

Although our study demonstrates the potential of energy-guided reranking, it has limitations. Energy estimates were model-based rather than derived from hardware measurements, introducing uncertainty for small differences. Experiments were confined to Python tasks on a single Apple Silicon system, and correctness validation relied on small inputs without fully controlling for caching or system load.

Future work will extend evaluation to larger, more diverse benchmarks, additional programming languages, and varied hardware platforms to assess generalizability. Hardware-level validation and reporting uncertainty through confidence intervals will enhance robustness, while improved correctness checks and system-level controls will strengthen rigor. Ultimately, integrating energy awareness directly into LLM training and developer tools could make sustainable, green-by-design software the norm rather than the exception.

## AI Agent Setup

We employed a dual-agent framework integrating ChatGPT-5 and Gemini 2.5 Pro to support multi-phase research planning, drafting, and refinement. ChatGPT-5 served as the primary agent for hypothesis generation, research planning, and manuscript writing, while Gemini 2.5 Pro was used for cross-validation and content refinement. The process began with domain-specific keywords provided to ChatGPT-5, which generated multiple hypotheses. These were evaluated by both human researchers and Gemini 2.5 Pro, and the most promising direction was selected based on combined feedback. ChatGPT-5 then elaborated the research plan and generated drafts for the Introduction, Methodology, and Experiment sections. For the Related Work section, we leveraged the Deep Research functionality of ChatGPT-5 to retrieve and summarize state-of-the-art literature. All references were manually verified, and invalid citations were iteratively corrected through additional GPT-5 queries. As part of the Methodology development, we also utilized Gemini's image generation capabilities to produce a study diagram that visually summarizes the workflow. Implementation was conducted in sequential phases, with AI agents focusing on one stage at a time. The Results, Discussion, and Conclusion sections were generated by ChatGPT-5 based on validated outputs. After receiving reviewer feedback, we provided the comments to ChatGPT-5 to assist in generating the camera-ready version, ensuring alignment with reviewer expectations and improving clarity and completeness.

## Acknowledgments and Disclosure of Funding

J.M. Aragón-Jurado would like to acknowledge the Spanish Ministerio de Ciencia, Innovación y Universidades for the support through FPU21/02026 grant.

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

# Agents4Science AI Involvement Checklist

1. **Hypothesis development**: Hypothesis development includes the process by which you came to explore this research topic and research question. This can involve the background research performed by either researchers or by AI. This can also involve whether the idea was proposed by researchers or by AI.

   Answer: [D]

   Explanation: Human researchers initially defined the research scope by providing key domain-specific keywords. AI tools then played a central role in generating multiple feasible hypotheses related to these keywords, leveraging prior literature and model knowledge. Researchers reviewed the AI-generated hypotheses, offering feedback and selecting the most promising direction. Once the direction was chosen, AI further elaborated the hypothesis details, including mechanisms, assumptions, and evaluation strategies, with iterative human feedback. This process was primarily AI-driven, with researchers curating and refining outputs to ensure relevance and novelty.

2. **Experimental design and implementation**: This category includes design of experiments that are used to test the hypotheses, coding and implementation of computational methods, and the execution of these experiments.

   Answer: [D]

   Explanation: After receiving the AI-generated research plan, the design of experiments and implementation of code were primarily performed by AI agents. These agents proposed experimental setups and generated executable code aligned with the selected hypothesis. Human researchers provided prompts, executed the experiments based on AI outputs, monitored runtime behavior, offered iterative feedback, and performed final cleanup—removing unused files. Throughout the process, researchers supervised execution quality and ensured scientific validity, while AI handled the generation and automation of core methodological components.

3. **Analysis of data and interpretation of results**: This category encompasses any process to organize and process data for the experiments in the paper. It also includes interpretations of the results of the study.

   Answer: [D]

   Explanation: Result processing and analysis were entirely conducted by AI agents. Human researchers provided prompts and high-level guidance to shape the evaluation scope, but did not intervene in the data handling or interpretation. Researchers reviewed outputs only at a strategic level to ensure alignment with research goals.

4. **Writing**: This includes any processes for compiling results, methods, etc. into the final paper form. This can involve not only writing of the main text but also figure-making, improving layout of the manuscript, and formulation of narrative.

   Answer: [D]

   Explanation: The main content of the paper—including text, figures, and narrative structure—was primarily generated by AI agents. Researchers provided prompts and high-level guidance to shape the overall direction and ensure consistency with experimental execution. Feedback from researchers focused on specific implementation details and formatting corrections to align the manuscript with actual procedures and submission requirements.

5. **Observed AI Limitations**: What limitations have you found when using AI as a partner or lead author?

   Description: AI performs well in generating ideas, drafting content, and executing workflows under human guidance. However, in our experience, different models tend to vary in output quality across tasks. GPT-5 appeared to perform better in planning and contextual understanding, while Gemini 2.5 Pro seemed more reliable for code generation. Based on these observed tendencies, we assigned GPT-5 as the lead author and used Gemini 2.5 Pro as a supporting agent for code correction and content refinement. Additionally, AI-generated outputs often require manual correction to ensure data and content consistency and adherence to formatting standards. Prompt design and high-level guidance from researchers remain essential to uphold scientific rigor.

